# Hub Gene Mining and Co-Expression Network Construction of Low-Temperature Response in Maize of Seedling by WGCNA

**DOI:** 10.3390/genes14081598

**Published:** 2023-08-07

**Authors:** Tao Yu, Jianguo Zhang, Jingsheng Cao, Xuena Ma, Wenyue Li, Gengbin Yang

**Affiliations:** 1Maize Research Institute of Heilongjiang Academy of Agricultural Sciences, Harbin 150086, China; zhangjianguo72@163.com (J.Z.); caoj.s@163.com (J.C.);; 2Key Laboratory of Biology and Genetics Improvement of Maize in Northern Northeast Region, Ministry of Agriculture and Rural Affairs, Harbin 150086, China; 3Key Laboratory of Germplasm Resources Creation and Utilization of Maize, Harbin 150086, China

**Keywords:** maize, low-temperature stress, weighted gene co-expression network, transcriptome, the seedling stage

## Abstract

Weighted gene co-expression network analysis (WGCNA) is a research method in systematic biology. It is widely used to identify gene modules related to target traits in multi-sample transcriptome data. In order to further explore the molecular mechanism of maize response to low-temperature stress at the seedling stage, B144 (cold stress tolerant) and Q319 (cold stress sensitive) provided by the Maize Research Institute of Heilongjiang Academy of Agricultural Sciences were used as experimental materials, and both inbred lines were treated with 5 °C for 0 h, 12 h, and 24 h, with the untreated material as a control. Eighteen leaf samples were used for transcriptome sequencing, with three biological replicates. Based on the above transcriptome data, co-expression networks of weighted genes associated with low-temperature-tolerance traits were constructed by WGCNA. Twelve gene modules significantly related to low-temperature tolerance at the seedling stage were obtained, and a number of hub genes involved in low-temperature stress regulation pathways were discovered from the four modules with the highest correlation with target traits. These results provide clues for further study on the molecular genetic mechanisms of low-temperature tolerance in maize at the seedling stage.

## 1. Introduction

Various abiotic stresses, such as low temperature, drought, and high salinity, are important factors affecting the normal growth and yield of plant. Among them, low-temperature stress has an influence on the growth, development, yield and spatial distribution of plant. As a typical cold-sensitive crop, maize (*Zea mays* L.) is one of the most important grain, feed and biomass energy sources in China, and the planting area reached 41.3 million hectares in 2019 [1,2]. As the largest province in maize cultivation and production in China, Heilongjiang Province accounts for approximately 15% of total maize planting area [3]. However, due to its unique geographical environment, low-temperature damage occurs frequently in spring in Heilongjiang Province, causing varying degrees of damage to seed germination and seedling growth of maize. It has been reported that a 1 °C drop in temperature may delay the maturity period by 10 days and reduce the yield by more than 10% [4]. Additionally, low temperatures will slow down the rate of leaf emergence, which may reduce the total number of leaves [5,6]. Researches have shown that low temperatures can even reduce the yield of maize by more than 15% in Heilongjiang province [7], seriously affecting the production security of maize. Therefore, a deeper understanding of the response mechanism of maize to low-temperature stress, and the mining of response genes, will provide an important theoretical basis for analyzing the adaptation of maize to low-temperature environments and creating new varieties with high quality and low-temperature tolerance.

Weighted gene co-expression networks analysis (WGCNA) cluster genes with similar expression patterns in the same module based on the correlation between gene expression was obtained from high-throughput sequencing technology, using the characteristics of the interrelatedness of life activities in plant. WGCNA analyzes the power system of connections between genes. It makes the genes within the module conform to the scale-free network topology. In the network, a few genes are connected to the majority of genes. By analyzing their correlations, it is possible to predict the central genes, which may be the key regulatory genes of the network [8]. Compared to traditional two-sample comparative analysis, the WGCNA method can efficiently handle multi-sample data processing, and is therefore widely used to study the biological relationship between co-expression networks and plant traits, as well as to identify key genes highly associated with traits. Since the weighted gene co-expression network analysis was proposed by Langfelder and Horvath in 2008, many key genes related to plant phenotype traits, responses to biotic or abiotic stress, and other aspects have been identified through this method [9]. Kuang et al. [10] used banana fruit as the research object and constructed a transcription factor regulatory network that regulates banana fruit maturation based on WGCNA, obtaining 25 key transcription factors involved in fruit maturation by regulating the expression of downstream maturation-related genes. Sun et al. [11] revealed the modules associated with green pigment accumulation through WGCNA by identifying differentially expressed genes between green and white fiber cotton, and identified 56 core genes, including two homologous genes of *Gh4CL4*, which participates in the biogenesis of green pigment. Zou et al. [12] identified a total of five specific modules related to fiber development by WGCNA on fiber transcriptome data of two cotton lines at different developmental stages, and excavated the hub genes in the modules. Greenham et al. [13] used the WGCNA method to analyze the transcriptome changes in Brassica crops in the early stage of drought stress response, and identified six hub genes for drought stress resistance, including cell response regulator 3 (*CRR3*), plastid-specific ribosomal protein 6 (*PSRP6*), and auxin polar output vector 3 (*PIN3*). Tan et al. [14] identified 22 gene modules by analyzing transcriptome data of 17 rice (*Oryza sativa* L.) at different time points treated with cadmium, and combined with differential expression analysis, a total of 164 genes related to cadmium stress response were mined. Ma et al. [15] used transcriptome data of different varieties of maize at two planting densities, combined with the WGCNA method to construct co-expression networks under the two density conditions, identified 15 co-expression modules with significant and highly correlated plant height and ear height, including 6 modules with the same two traits, with the reported plant height gene as the core, constructed a gene network, and found auxin transcription factors ARFTF7, ARFTF26, GST39, photosynthetic system II oxygen evolution polypeptide PspB2 and photosynthetic system IN subunit PasN1 is associated with core genes. By performing WGCNA on 14 transcriptome data at different developmental stages of maize, the researchers identified 14 tissue-specific modules, and further studied the gene interaction network of 2 of them, from which hub genes such as *ZCN8*, *ZCN7*, *COL1* and other flowering-related hub genes were mined [16].

In this study, differential expression analysis was conducted on the transcriptome data maize leaves treated at 5 °C at different time points, WGCNA was used to construct gene co-expression networks, associated gene expression modules with low-temperature treatment, explored module functions via GO enrichment analysis, and identified and constructed a co-expression network of core genes with differential expression during maize seedling resistance to low-temperature stress, in order to provide new clues and ideas for further study on the molecular mechanisms of maize tolerance to low-temperature stress.

## 2. Materials and Methods

### 2.1. Experimental Materials and Treatments

The experimental materials of maize inbred lines B144 (cold stress tolerant) and Q319 (cold stress sensitive) were provided by the Maize Research Institute of Heilongjiang Academy of Agricultural Sciences, China. Seeds were germinated in a 25 °C light incubator (GEN1000, Conviron, Pembina, ND, USA) under 12/12 h light/dark cycle with a mixed substrate of nutrient soil and vermiculite (volume ratio of 3:1) for 10 days, ensuring adequate moisture. Then, the temperature was dropped to 5 °C to induce a low-temperature stress. The leaf samples were taken after 0 h, 12 h and 24 h from both inbred lines in three replicates. The sequencing results were uploaded to NCBI (https://www.ncbi.nlm.nih.gov/bioproject/PRJNA666026, accessed on 28 September 2020 [17]). The treatment groups were named BCK, B12, B24 for B144 and QCK, Q12, Q24 for Q319 according to the treatment time (0, 12, 24 h).

### 2.2. Clustering Analysis

After filtering the required genes using the function of goodSamplesGenes provided by the R package WGCNA, weighted gene co-expression network analysis (WGCNA) [9] was performed using the R package to obtain a more accurate co-expression network. The soft-thresholding power was determined based on the principle of scale-free networks, and the software-provided soft-thresholding power was used for subsequent analysis. Dynamic tree-cutting method was used to identify co-expression patterns and to construct gene clustering trees based on the correlation of gene expression levels. The minimum number of genes in a module was set to 30, and modules with similar expression patterns were merged based on a similarity threshold of 0.75 for module eigengenes. The relationship between the quality traits and each module was analyzed, and the absolute value of the correlation between a certain trait and a certain module approaching 1 indicates a stronger correlation between the genes in that module and that trait. Heat maps were drawn to identify the modules significantly correlated with specific samples for further analysis.

### 2.3. Functional Enrichment of Module Genes

The R package clusterProfiler was used for GO (Gene Ontology) functional analysis and KEGG (Kyoto Encyclopedia of Genes and Genomes) pathway analysis of the genes in the selected modules [18]. GO terms and KEGG pathways with *p* value ≤ 0.05 were considered differentially enriched.

### 2.4. Construction of Gene Co-Expression Networks

The gene co-expression network output by WGCNA was processed and the hub genes were screened using Cytoscape_3.7.2 [19]. Each node in the network represents a gene, and the edges represent the relationship between genes. The gene co-expression network accurately identifies potential hub genes and predicts the functions of unknown genes by using the known functions of other genes. All analyses in this study were based on R 3.6.0. Some figures and plots were drawn using the R packages ggplot2 and pheatmap.

## 3. Results and Analysis

### 3.1. RNA Sequencing

The transcriptome sequencing with Illumina HiseqTM 4000 sequencing platform resulted in 113.56 million raw reads from 18 samples. After filtering, a total of 149.03 Gb clean reads was obtained with Q30 based percentage greater than 93.30% and GC percent ranging from 54.02% to 57.82%. Among the 18 samples, the matching rate of clean reads to the reference genome was greater than 85%, indicating that all experimental samples were highly reliable in terms of collection and sequencing results.

To identify differentially expressed genes (DEGs) between different treatment comparisons, DESeq2 software was used to screen the DEGs with the log_2_ FC (log_2_ fold change) ≥ 2, padj < 0.01. After 12 h of low-temperature treatment, there were 5824 DEGs in B144, of which 3401 genes were up-regulated and 2423 genes were down-regulated; there were 7446 DEGs in Q319, of which 4189 genes were up-regulated and 3257 genes were down-regulated. After 24 h of low-temperature treatment, there were 4929 DEGs in B144, of which 2524 genes were up-regulated and 2405 genes were down-regulated; there were 430 DEGs in Q319, of which 389 genes were up-regulated and 41 genes were down-regulated. The above results show that with the prolongation of low-temperature stress time, B144 induced more DEGs compared to Q319, which were used to regulate changes in metabolic pathways of the plant to defend against low-temperature damage.

### 3.2. Determination of Soft Threshold in Gene Co-Expression Networks

In order to make the co-expression networks follow the scale-free networks distribution, the function pickSoftThreshold in the R package WGCNA was used to calculate the weight value and selected the appropriate soft threshold. The result showed that the optimal soft threshold β = 5 (Figure 1A), and the network connectivity under different soft thresholds was shown in Figure 1B, which will be further used to construct the co-expression networks.

### 3.3. Gene Clustering and Module Segmentation in Gene Co-Expression Networks

After determining the soft threshold β = 5, the similarity matrix was transformed into an adjacency matrix, and then the adjacency matrix was transformed into a topological overlap matrix (TOM). In order to accurately transform the topological matrix into an dissimilarity matrix, the formula dissTom = 1-TOM was used to clear the errors caused by background noise and pseudo-associations. Finally, the hierarchical clustering generated by the function hcluster was segmented by dynamic cutting. Genes in the same branch had similar expression patterns, and each branch represented a co-expression module, which was distinguished by different colors. Genes that could not be assigned to any module were displayed in gray. The differentially expressed genes were clustered based on their expression levels, and modules with feature vector values below 0.2 were merged. Finally, 33 modules were obtained. Among them, the grey module contained 10 genes that could not be assigned to any module, the turquoise module contained the most genes, with 10,898 genes, and the violet module contained the fewest genes, with 39 genes (Figure 2).

### 3.4. Identification of Specific Modules under Low-Temperature Stress

The modules were associated with the low-temperature treatment samples, and 12 modules were highly correlated with low-temperature traits (|r| > 0.70, *p* < 0.001). Among them, the B144 of 12 h was highly positively correlated with the cyan, orange, and salmon modules (r = 0.98, *p* = 3 × 10^−13^; r = 0.99, *p* = 8 × 10^−16^; and r = 0.98, *p* = 3 × 10^−12^, respectively); the B144 of 24 h was highly positively correlated with the darkgrey, tan, and midnightblue modules (r = 0.99, *p* = 4 × 10^−14^; r = 0.99, *p* = 1 × 10^−14^; and r = 0.99, *p* = 4 × 10^−15^, respectively); the Q319 of 12 h was highly positively correlated with the darkorange, purple, and lightyellow modules (r = 0.99, *p* = 4 × 10^−16^; r = 0.99, *p* = 3 × 10^−16^; and r = 0.99, *p* = 8 × 10^−15^, respectively); and the Q319 of 24 h was highly positively correlated with the greenyellow, magenta, and royalblue modules (r = 0.99, *p* = 8 × 10^−17^; r = 1, *p* = 7 × 10^−18^; and r = 0.99, *p* = 1 × 10^−16^, respectively) (Figure 3).

The feature vector genes (module eigengene, ME) of the 33 modules were clustered, and by comparing the correlation between MEs and combining the results of module correlation with low-temperature traits, it was found that the correlation between the MEs of the darkorange and greenyellow modules was 0.99, and the correlation between the MEs of the lightyellow and purple modules was 0.99, and the correlations between these four modules in B144 and Q319 were opposite (Figure 4). Finally, it was determined that darkorange module, greenyellow module, lightyellow module and purple module were specific modules for low temperatures of maize.

### 3.5. Enrichment Analysis of Specific Modules under Low-Temperature Stress

In order to explore the gene functions within the specific modules under low-temperature stress, the R package clusterProfiler was used to perform GO annotation on the four identified modules, darkorange, greenyellow, lightyellow and purple (Figure 5).

The genes in the darkorange module were mainly enriched in cellular components such as Chloroplast (GO:0009507), transcription factor TFIIH holo complex (GO:0005675), anaphase-promoting complex (GO:0005680) and pre-ribosome (GO:0030688); biological processes including RNA modification (GO:0009451), positive regulation of transcription by RNA polymerase II (GO:0045944) and transcription by RNA polymerase II (GO:0006366); and molecular functions such as protein binding (GO:0005515), transferase activity (GO:0016747), 3′-5′ exonuclease activity (GO:0008408) and NADH dehydrogenase activity (GO:0003954).

The genes in the greenyellow module were mainly enriched in cellular components such as cell wall (GO:0005618) and plant cell wall (GO:0009505); biological processes including GTPase activity activation (GO:0090630), and mitotic cell cycle phase transition (GO:0044772); cyclin-dependent protein serine/threonine kinase regulatory activity (GO:0016538), GTPase activator activity (GO:0005096) and other pathways of molecular function.

The genes in the lightyellow module were mainly enriched in cellular components such as cell wall of cellular fractions (GO:0005618), the PP2A complex (GO:0000159), and the GTPase activator activity (GO:0005096); biological processes including transcriptional regulation of biological processes (GO:0045893), and distal axis cell differentiation (GO:0010158); molecular function of protein phosphatase regulator activity (GO:0019888), DNA-dependent ATPase activity (GO:0008094), structural composition of nuclear pores (GO:0017056) and other pathways.

The genes in the purple module were mainly enriched in cellular components such as plant-type cell wall (GO:0009505), plasmodesma (GO:0009506), extracellular space (GO:0005615) and extracellular region (GO:0005576); biological processes including response to stress (GO:0006950), xyloglucan biosynthetic process (GO:0009969), mitotic cell cycle phase transition (GO:0044772) and regulation of cyclin-dependent protein serine/threonine kinase activity (GO:0000079); molecular function of peroxidase activity (GO:0004601), chitinase activity (GO:0004568) and cyclin-dependent protein serine/threonine kinase regulator activity (GO:0016538).

KEGG annotation of four specific modules, darkorange, greenyellow, lightyellow and purple, was performed using the KOBAS online tool. The results showed that the genes within the modules were highly significantly enriched in plant MAPK signaling pathway, plant hormone signal transduction, metabolic pathway, ribosome, biosynthesis of secondary metabolites, biosynthesis of amino acids, and basal transcription factors (Figure 6).

### 3.6. Construction of Gene Co-Expression Networks

Hub genes usually refer to genes with high connectivity within a module. In this study, we selected the top 10 genes with the highest connectivity in the darkorange, greenyellow, lightyellow, and purple modules as the hub genes. The Cytoscape software 3 6.1 was used to visualize the hub genes and their associated genes, and to construct a gene interaction network diagram (Figure 7). In these networks, each node represents a gene, and nodes are connected by edges, and genes at the ends of the edges are usually considered to have the same biological function. In order to obtain the functional information of these hub genes, we used the NCBI database (https://www.ncbi.nlm.nih.gov/, accessed on 28 September 2020) to query the relevant information of these hub genes in rice, and annotated their functions (Table 1).

## 4. Discussion

WGCNA analysis provides new ideas and methods for studying gene regulatory networks of different traits, and it has been widely used in various scientific fields. In this study, four specific modules (darkorange, greenyellow, lightyellow, and purple) related to low-temperature tolerance of maize at the seedling stage were identified through the WGCNA method. By calculating the connectivity of the characteristic gene in the modules, the importance of the gene in the network can be inferred. The 10 genes with the highest connectivity in each specific module were selected as the hub genes, and it was speculated that they might play an important role in the low-temperature tolerance of maize at the seedling stage.

Among the top 10 hub genes in the darkorange module, *LOC103646333* annotated a heavy metal-associated isoprenylated plant protein (HIPP), a metal chaperone molecule that plays a key role in plant physiological activities, mainly in binding and translocating metal ions to target proteins to maintain metal ion homeostasis in the cell. *OsHIPP41* was found to be highly expressed in response to cold and drought stress, and its product was located in the cytoplasm and nucleus. The results suggest that HIPPs play an important role in vascular plant development and plant responses to environmental changes [20]. Cui et al. [21] used CRISPR/Cas9 system to create a *GmHIPP26* gene mutant and studied its function in cadmium stress, and the results showed that the *GmHIPP26* gene plays an important role in alleviating plant cadmium stress and cadmium transport in plants. These results suggested that HIPPs play an important role in plant development and response to environmental changes, including responses to environmental stresses such as heavy metals excess, cold and drought, as well as plant-pathogen interactions. *LOC103635071* encodes an E3 ubiquitin protein ligase, which participates in a variety of physiological processes within cells by regulating the ubiquitination process of proteins, and plays an important role in regulating plant responses to abiotic stress. Choi et al. [22] found that *OsCBE1* encodes a novel substrate receptor for Cullin4-Based E3 ubiquitin ligase complex (C4E3), and *OsCBE1* is involved in the regulation of development and abiotic stress response. Meanwhile, the mutant of *OsCBE1* was insensitive to ABA during seed germination, indicating that *OsCBE1* participated in the stress response through ABA signaling pathway. *OsCBE1*, as a member of C4E3, plays a regulatory role in abiotic stress responses via CCCH. Li et al. [23] demonstrated the role of XBAT35.2 in the abiotic stress tolerance of Arabidopsis. The loss of E3 ubiquitin ligase activity reduced the sensitivity of seedlings to salt stress, while the overexpression of E3 ubiquitin ligase activity made seedlings more sensitive to salt stress, suggesting that E3 ubiquitin ligase functions as a negative regulator of tolerance. There is evidence that XBAT35.2 promotes protease-dependent degradation of ACD11 during abiotic stress to attenuate the response. This coincides with the finding that ACD11 promotes tolerance to salt and drought stresses. Cui et al. [24] found that *OsATL38* was identified as a low-temperature-induced gene that negatively regulates the cold stress response in rice via mono-ubiquitination of *OsGF14d 14-3-3* protein. Kim et al. [25] suggested that *AtATL78* was a negative regulator of cold response and a positive regulator of drought response, meanwhile, *AtATL78* played opposing roles in cold and drought stress responses. These results suggest that E3 ubiquitin ligase is involved in regulating responses to biotic and abiotic stress.

Among the top 10 hub genes of the greenyellow module, *LOC103639446* encodes a plant calmodulin-binding protein (CaMBP), the most important class of Ca^2+^-sensing proteins that regulate cellular physiological functions by interacting with its calmodulin-binding proteins. The expression of CaMBPs has been shown to be induced by various abiotic stresses, biotic stresses, salicylic acid (SA), ethylene and jasmonic acid (JA) hormones, regulating downstream target genes and participating in hormone signalling, ion transport, gene transcription and other pathways, thus playing a key role in regulating the plant’s response to stress. CaMBP was first identified in Arabidopsis, and CBP60s have an important role in the immune response of plants. Studies have shown that CBP60g is a positive regulator of plant immunity, and overexpression of CBP60g promotes SA accumulation in Arabidopsis, while inducing the expression of disease-stage related genes and ICS1 genes to improve disease resistance [26]. Lv et al. [27] found that *AtIQM1* was a Ca21-independent CaMBP, indicated that IQM1 was a key regulatory factor in signaling of plant disease responses mediated by JA, indicating CaMBP may play a critical role in the cross talk of multiple signaling pathways of plant defense processes. The CBP60 family is also involved in regulating a variety of abiotic stresses in plants [28]. The results of Zhang Xin’s study showed that Arabidopsis AtCBP60g plays a role in resistance to biotic stresses and mediates phytohormone and abiotic stress signaling pathways [29,30]. Recently, CAMTA3 has been shown to positively regulate plant responses to drought stress [31]. In Arabidopsis, the CBF cold response pathway has been shown to regulate plant tolerance to low temperature, and CAMTA3 is a positive regulator of CBF2 expression [32]. *LOC100191502* encodes a serine carboxypeptidase-like protein (SCPL). Studies have shown that SCPL genes play a role in the hydrolysis of storage proteins during seed germination, programmed cell death (PCD), seed development, stress resistance and many other processes, while 54 SCPL genes have been identified in Arabidopsis thaliana and many have been cloned. Wang et al. [33] identify the whole genome of cotton SCPL genes and indicated that *GhSCPL42* gene was a positive regulator gene that played an important role in resistance to Verticillium wilt in cotton. Xu et al. [34] found that the *TaSCPL184-6D* gene enhanced transgenic *Arabidopsis* plant drought and salt tolerance. LOC100283475 was annotated as the peptidyl-prolyl cis-trans isomerase CYP19-4. In rice, *OsCYP19-4* had the activity of peptidyl-prolyl cis-trans isomerase, the cold tolerance of *OsCYP19-4* overexpression plants was enhanced, the number of tillers and panicles increased significantly, and the yield of rice increased. *OsCYP19-4* is secreted from the endoplasmic reticulum to the apoplasts via vesicle transport and is involved in the adaptive development to environmental stresses, particularly cold stress [35,36].

Among the top 10 hub genes of the lightyellow module, *LOC100381507* encodes peroxidase, a class of simple organelles that play an important role in plant reactive oxygen species (ROS) metabolism. Previous studies have revealed that ROS is involved in programmed cell death (PCD) in plants. PCD in plants is essential for the regulation of plant growth and development, and environmental stress tolerance [37,38,39]. *LOC100383295* encodes cytochrome P450. Plant cytochrome P450 (CYP450) is a monooxygenase encoded by the supergene family, known as a “universal biocatalyst”, once activated, can participate in a series of catalytic reactions and play an important role in signaling, biological defense, abiotic stress, and the synthesis and degradation of metabolites. At present, a large number of plant CYP450 genes have been identified. Studies have shown that the plant CYP450 gene family can also participate in the synthesis and degradation of plant endogenous hormones, thereby regulating the response of plant to stress. For example, the expression of *CYP94B1*, *CYP94B3* and *CYP94C1* in Arabidopsis Thaliana affects the jasmonic acid metabolic pathway, activates the expression of downstream stress resistance genes, and thus affects the stress response of plants [40]. Under drought stress, ABA hydroxylase genes *CYP707A1*, *CYP94C1* and *CYP94B3* in tobacco were significantly up-regulated, indicating that the expression of *CYP450* was induced by drought stress [41]. Studies have reported that *CYP86A1* in island cotton encodes cytochrome P450 fatty acid W-hydroxylase, which is a key enzyme in cork resin biosynthesis, and it has been found that the silencing of *CYP86A1* leads to a serious weakening of the resistance of island cotton to verticillium wilt, and the heterologous expression of *CYP86A1* in Arabidopsis thaliana improved the tolerance of Arabidopsis to verticillium wilt, and transcriptomic analysis of the overexpression of this gene not only affects the synthesis of root lipid substances, but also activates the disease-fighting immune system [42].

Among the top 10 hub genes of the purple module, 4 hub genes have the same function. *LOC100037802*, *LOC541856*, *LOC100037804* and *LOC100037810* belong to lipoxygenase (LOX). Studies have shown that lipoxygenase has a wide range of roles in plant growth and development, stress response and many other processes, mechanical injury, drought and other abiotic stress can induce the expression of *LOX* gene [43]. For example, four members of *13-LOX* (*LOX2*, *3*, *4*, and *6*) in Arabidopsis thaliana accelerate the synthesis of JA in leaves when induced by mechanical injury, and JA also plays an important role in plant response to mechanical injury [44]. The expression of *LOX3* in Arabidopsis thaliana is strongly induced by salt treatment [45]. At the same time, lipoxygenase also plays an important role in abiotic stresses and resisting biological stress. Shaban et al. [46] found that many cis-acting elements related to growth, stresses, and phytohormone signaling were found in the *GhLOX* genes. Upadhyay et al. [47] determined the transcript abundance patterns of 14 LOX genes in response to four independent abiotic stresses ( heat, cold, drought and salt ) and found that these LOX genes play important roles in abiotic stress in tamato. The transcriptional expression of *ZmLOX12* in maize was strongly induced by Verticillium wilt infection, its loss-of-function mutant lox12-1 was less resistant to pathogens, and the results showed that LOX gene mediated the defense response of maize to Verticillium wilt by positively regulating the synthesis of jasmonin and others [48]. *LOC100285766* annotated auxin amino acid hydrolase (ILL), an enzyme that catalyzes the release of free auxin by combining auxin amino acids, and regulating numerous developmental processes in plants by regulating free auxin. *LOC103641407* encodes a tryptophan-aspartate repeat sequence protein (WD40), which has been shown to be widely present in plants and involved in the regulation of numerous metabolic reactions. In recent years, more and more studies have reported the important role of WD40 protein in regulating plant growth and development under adverse conditions such as high salinity, drought and low temperature. Zhu et al. reported that a WD40 protein of Arabidopsis, *HOS15* (highly expressed osmotic stress response gene 15), is deacetylated by chromosomal histones and is critical for the inhibition of genes associated with tolerance to abiotic stress [49]. Ananieva et al. found that 5PTases13 in Arabidopsis thaliana can interact with and regulate its activity with SnRK1 through its WD repeat domain, SnRK1 is the integration center of plant metabolic response, stress, growth and development signals, and WD40 regulates plant stress signals by cross-integration between signaling pathways [50]. It has been reported that WD40 can also regulate osmotic stress via ubiquitination pathway and gene transcription [51]. Hu et al. [52] identified a total of 743 WD40 protein members in wheat genome, based on the result of RNA-seq data analysis, numerous *TaWD40s* were involved in responses to stresses, including cold, heat, drought, and powdery mildew infection pathogen. *LOC541838* encodes glutathione S-transferase, which has been shown to be a multifunctional proteases that protects the activity of cells and protein in plant and improves the resistance of plant under stress. Mahfuzur et al. cultivated the seedlings of cold-tolerant maize BARI hybrid maize-7 in a 4 °C environment. Compared with the control, the activity of GST increased with the prolongation of low-temperature stress time. Western blotting analysis showed that the expression of GST in cold-tolerant maize could reduce the oxidative damage caused by H_2_O_2_ to the cell structure of maize and maintain the growth of plants under cold stress [53]. Kumar et al. found that *OsGSTL2* has a positive effect on resisting cold stress. The results showed that *OsGSTL2* was tolerant to cold stress and other abiotic stresses such as heavy metals, osmotic stress and salt, and the germination rate, root length and GST activity of transgenic plants were higher than that of wild type. It is indicated that *OsGSTL2* improves the tolerance of plants to cold stress by enhancing the antioxidant system in Arabidopsis [54]. Duan et al. [55] identified 46 GST genes in cucumber, through the transcriptome and RT-qPCR analysis, it illustrates the characteristics and functions of *CsGST* genes and revealed that most *CsGST* members responded to cold stress in cucumber.

## 5. Conclusions

In this study, co-expression networks of weighted genes associated with low-temperature tolerance traits of maize were constructed, 12 gene modules significantly related to low-temperature tolerance at the seedling stage were obtained, then GO and KEGG analysis were carried out. Four modules with a high correlation degree with target traits were selected for in-depth analysis, among which 10 genes with the highest connectivity in each specific module were selected as the hub genes. Through functional annotation, it was found that some hub genes were closely related to the reported abiotic stress regulation pathway, and some functions were only studied in model plants. The results of this study provide clues for the study of the molecular mechanism of low-temperature tolerance in maize at the seedling stage, and provide theoretical support for the cultivation of new maize varieties of low-temperature tolerance.

## Figures and Tables

**Figure 1 genes-14-01598-f001:**
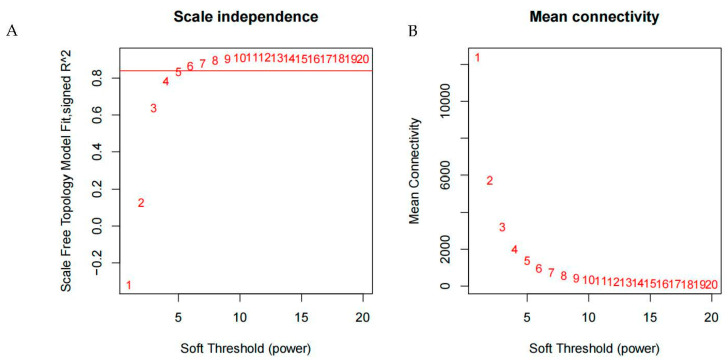
Soft threshold determination of gene co-expression network. Note: (**A**) The abscissae is the soft threshold (β) and the ordinate is the scale-free network model index. (**B**) The abscissae is the soft threshold (β) and the ordinate is the mean connectivity.

**Figure 2 genes-14-01598-f002:**
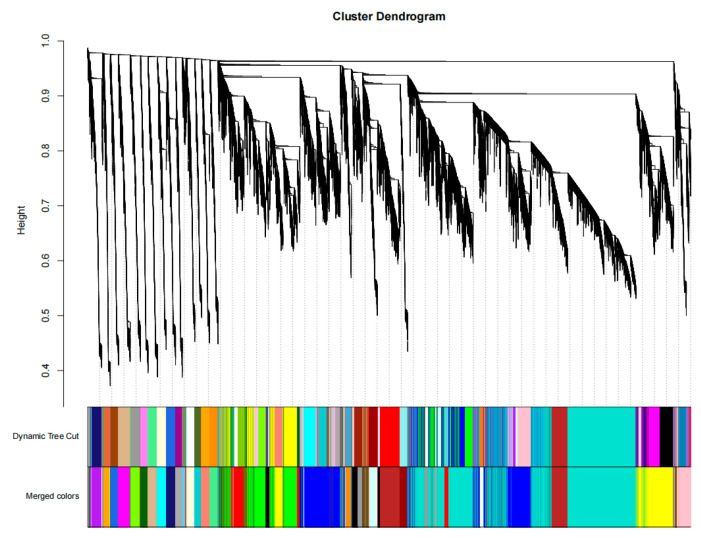
Gene clustering tree and module cutting.

**Figure 3 genes-14-01598-f003:**
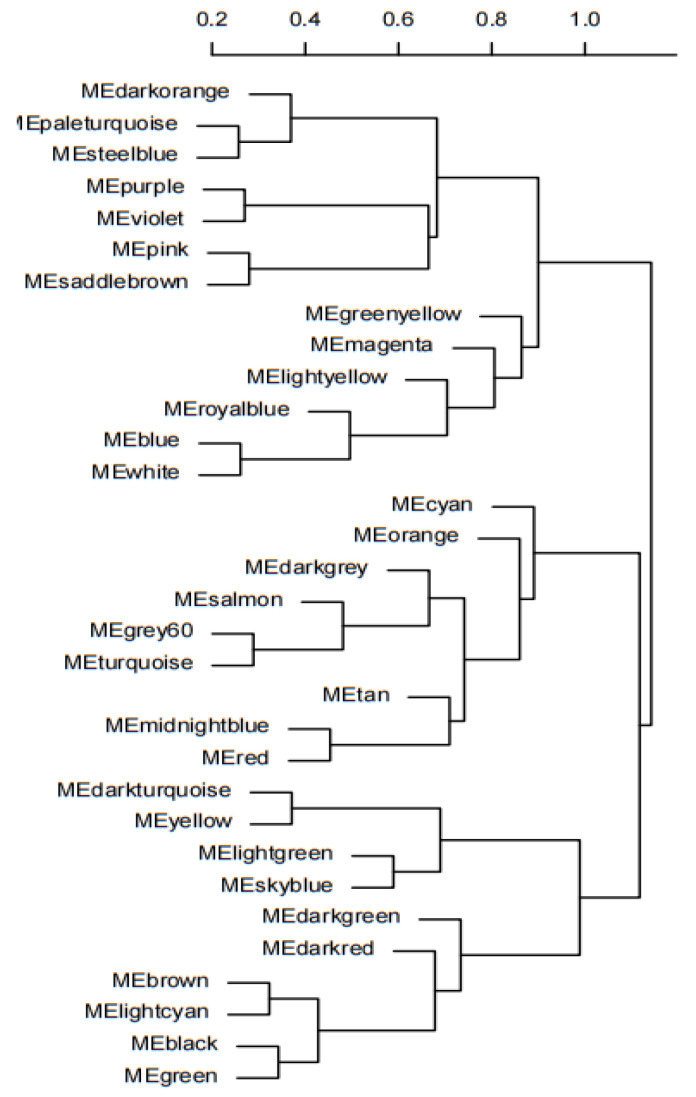
Association analysis of gene co-expression network modules with traits.

**Figure 4 genes-14-01598-f004:**
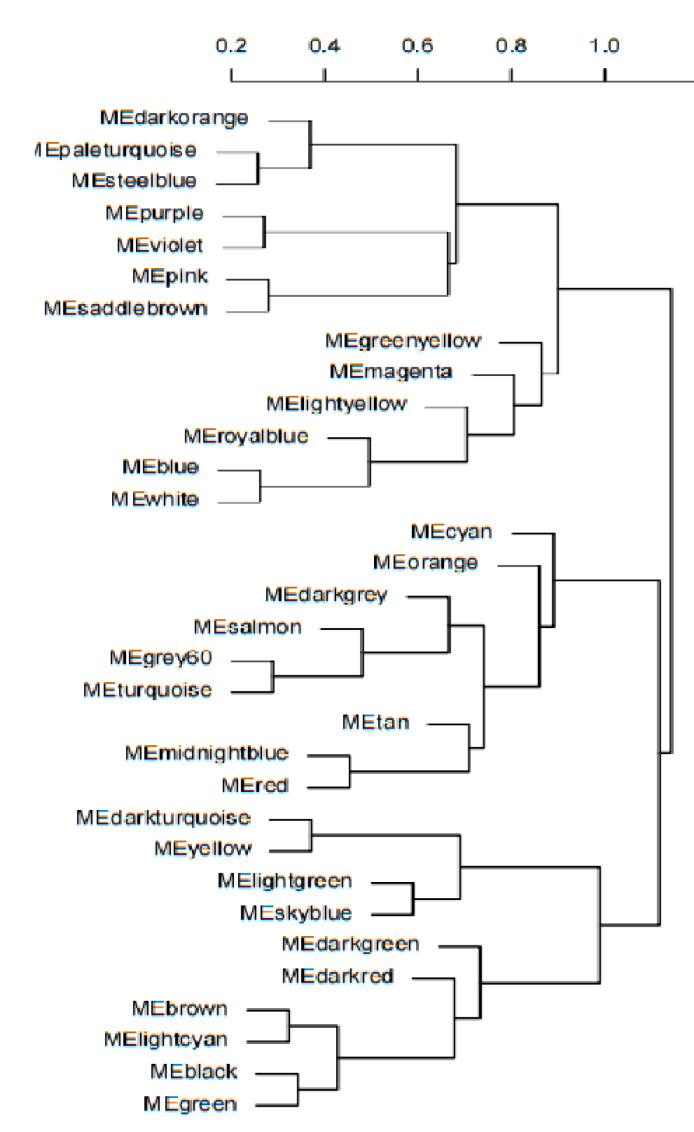
ME cluster tree.

**Figure 5 genes-14-01598-f005:**
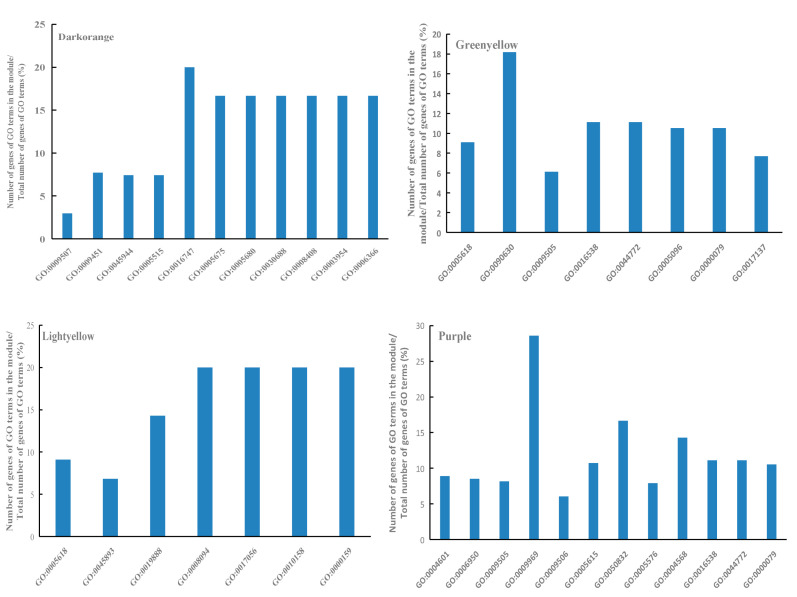
GO annotation of module genes.

**Figure 6 genes-14-01598-f006:**
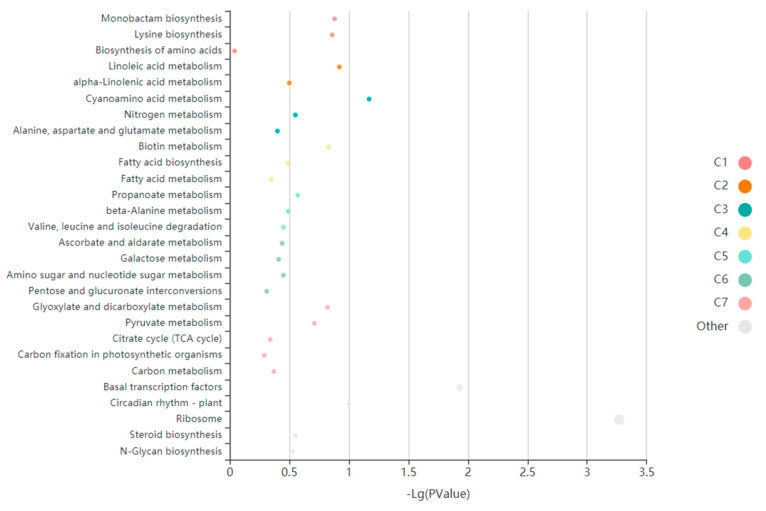
KEGGEnriched terms visualized in bubble plot. Each bubble represents an enriched function, and the size of the bubble from small to large: [0.05, 1], [0.01, 0.05), [0.001, 0.01), [0.0001, 0.001), [1 × 10^−10^, 0.0001), [0, 1 × 10^−10^). The color of the bar is the same as the color in the circular network, which represents different clusters. For each cluster, if there are more than five terms, the top five with the highest enrich ratio will be displayed.

**Figure 7 genes-14-01598-f007:**
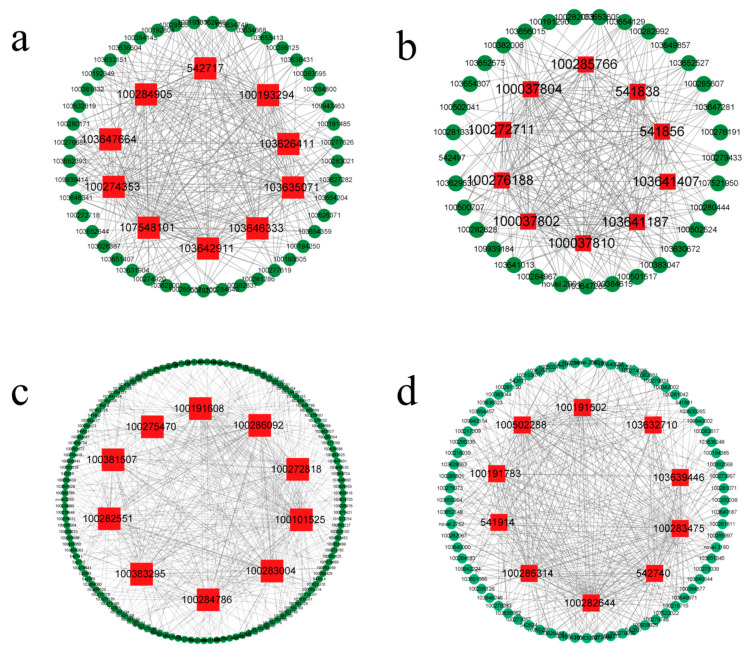
Gene co-expression networks of the specific modules. Note: (**a**) darkorange module network; (**b**) purple module network; (**c**) lightyellow module network; (**d**) greenyellow module network. Red represents the hub genes of the modules, and green represents the related genes of the hub genes.

**Table 1 genes-14-01598-t001:** Functional annotation of hub genes in the modules associating with low-temperature tolerance.

Module Name	Hub Gene	Homologous Gene in Rice	Gene Annotation in Maize
darkorange	LOC103646333	LOC4336080	heavy metal-associated isoprenylated plant protein 47
	LOC100193294	LOC4331934	sulfate transporter3
	LOC103647664	LOC4343201	argonaute protein 18
	LOC103626411	-	putative disease resistance protein RGA3
	LOC103635071	LOC107277552	E3 ubiquitin-protein ligase EL5
	LOC100284905	LOC4331734	uncharacterized
	LOC107548101	LOC4326262	uncharacterized
	LOC542717	LOC4326149	isoflavone reductase-like 1
	100274353	LOC9269286	uncharacterized
	103642911	LOC4346124	exocyst complex component EXO70B1
greenyellow	LOC541914	LOC4326376	aldehyde dehydrogenase 5
	LOC100502288	LOC4325649	lipid phosphate phosphatase 2
	LOC100191783	LOC4349708	carbohydrate transporter/sugar porter/transporter
	LOC103640644	LOC4352852	thaumatin-like protein
	LOC100286314	LOC4341938	phosphate import ATP-binding protein pstB 1
	LOC100283475	LOC4342022	peptidyl-prolyl cis-trans isomerase CYP19-4
	LOC103632710	LOC4346970	type IV inositol polyphosphate 5-phosphatase 9
	LOC103639446	LOC4332030	plant calmodulin-binding protein-related
	LOC100191502	LOC4347942	serine carboxypeptidase-like 19
	LOC542740	LOC4325704	glutathione transferase 8
lightyellow	LOC100381507	LOC4341247	peroxidase 52
	LOC100101525	LOC4327535	cysteine proteinase inhibitor
	LOC100286092	LOC4337884	uncharacterized
	LOC100272818	LOC4339940	aspartyl protease AED1
	LOC100383295	CYP714B2	cytochrome P450 714B3
	LOC100283004	LOC4334796	uncharacterized
	LOC100282551	LOC4336977	UDP-N-acetylglucosamine diphosphorylase
	LOC100275470	LOC4337397	uncharacterized
	LOC100191608	LOC4330180	HXXXD-type acyl-transferase family protein
	LOC100284786	-	uncharacterized
purple	LOC100037802	LOC4334049	lox2 linoleate 9S-lipoxygenase2
	LOC100285766	LOC4327677	IAA-amino acid hydrolase ILR1-like 4
	LOC541856	-	lox1 linoleate 9S-lipoxygenase1
	LOC100272711	-	uncharacterized
	LOC100037804	-	linoleate 9S-lipoxygenase5
	LOC100276188	-	uncharacterized
	LOC103641407	LOC4332648	putative WD40-like β propeller repeat family protein
	LOC103641187	LOC4344635	(E)-β-farnesene synthase-like
	LOC100037810	LOC4328603	lox11 linoleate 13S-lipoxygenase11
	LOC541838	LOC4347319	glutathione transferase25

## Data Availability

The data used are available within the text.

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
