# Peer review of "Hub Gene Mining and Co-Expression Network Construction of Low-Temperature Response in Maize of Seedling by WGCNA"

_genes, 2023, doi:10.3390/genes14081598_

Round 1
Reviewer 1 Report
Dear Authors,
Your work presented to me for review is very interesting. It was a great pleasure for me to read this work and review it. To be honest, a certain shortcoming of the work is quite a small number of cited scientific papers, but this is probably the result of a strict selection of the most important scientific articles. This is not a complaint and does not affect the quality and value of the work, but usually a minimum of 50 works of cited literature are used.
Good luck in your further research work.
Author Response
Response to Reviewer 1 Comments
The Reviewer 1: Your work presented to me for review is very interesting. It was a great pleasure for me to read this work and review it. To be honest, a certain shortcoming of the work is quite a small number of cited scientific papers, but this is probably the result of a strict selection of the most important scientific articles. This is not a complaint and does not affect the quality and value of the work, but usually a minimum of 50 works of cited literature are used.
Response : We thank the reviewer for his/her consideration that our work is very interesting. We have added some contents relevant to this research in the revised manuscript, and 52 papers were ultimately cited.
Reviewer 2 Report
The manuscript “Hub gene mining and co-expression networks construction of low temperature response in maize of seedling by WGCNA” in the abstract, introduction, and materials and methods sections, it looks like interesting research. However, since the figures were not included, which are key to this research, it is difficult to analyze the results and the discussion of the manuscript. Therefore, it is recommended that the authors include them in the manuscript; as well as to include in the introduction some statistical data about the importance of maize crop in their Country or Province, briefly describe the varying degrees of damage to seed germination and maize seedling growth caused by low temperatures, and specify the different time points at which plant material sampling was performed.
Author Response
The Reviewer 2: The manuscript “Hub gene mining and co-expression networks construction of low temperature response in maize of seedling by WGCNA” in the abstract, introduction, and materials and methods sections, it looks like interesting research. However, since the figures were not included, which are key to this research, it is difficult to analyze the results and the discussion of the manuscript. Therefore, it is recommended that the authors include them in the manuscript; as well as to include in the introduction some statistical data about the importance of maize crop in their Country or Province, briefly describe the varying degrees of damage to seed germination and maize seedling growth caused by low temperatures, and specify the different time points at which plant material sampling was performed.
Response : We thank the reviewer for his/her consideration that our work is an interesting research. We have modified the results and analysis as advised by the reviewer by adding some key figures of this research; we have modified the introduction as advised by the reviewer by adding some statistical data about the importance of maize crop in China and Heilongjing province; we have modified the introduction as advised by the reviewer by adding the damage to seed germination and maize seedling growth caused by low temperature; we have modified the materials and methods as advised by the reviewer by adding the different time points at which plant material sampling was performed.

Round 2
Reviewer 2 Report
The manuscript “Hub gene mining and co-expression networks construction of low temperature response in maize of seedling by WGCNA” looks like interesting research. However, again the images of the figures that are mentioned in the manuscript were not included. Therefore, it is recommended that the authors include them in the archives of the manuscript.
Author Response
The Reviewer 2: The manuscript “Hub gene mining and co-expression networks construction of low temperature response in maize of seedling by WGCNA” looks like interesting research. However, again the images of the figures that are mentioned in the manuscript were not included. Therefore, it is recommended that the authors include them in the archives of the manuscript.
Response : We thank the reviewer for his/her consideration that our work is an interesting research. We have modified the results and analysis as advised by the reviewer by adding the table and figures of this research in the manuscript.
